# Dose-Dependent Effects in Plasma Oncotherapy: Critical In Vivo Immune Responses Missed by In Vitro Studies

**DOI:** 10.3390/biom13040707

**Published:** 2023-04-21

**Authors:** Yuanyuan He, Fanwu Gong, Tao Jin, Qi Liu, Haopeng Fang, Yan Chen, Guomin Wang, Paul K. Chu, Zhengwei Wu, Kostya (Ken) Ostrikov

**Affiliations:** 1School of Nuclear Science and Technology, University of Science and Technology of China, Hefei 230026, China; 2Department of Geriatrics, Division of Life Sciences and Medicine, The First Affiliated Hospital of USTC, University of Science and Technology of China, Hefei 230001, China; 3Department of Medical Oncology, Division of Life Sciences and Medicine, The First Affiliated Hospital of USTC, University of Science and Technology of China, Hefei 230001, China; 4Joint Laboratory of Plasma Application Technology, Institute of Advanced Technology, University of Science and Technology of China, Hefei 230026, China; 5Department of Orthopedics, School of Medicine, Shanghai Tenth People’s Hospital, Tongji University, Shanghai 200072, China; 6Department of Physics, City University of Hong Kong, Tat Chee Avenue, Kowloon, Hong Kong 999077, China; 7Department of Materials Science and Engineering, City University of Hong Kong, Tat Chee Avenue, Kowloon, Hong Kong 999077, China; 8Department of Biomedical Engineering, City University of Hong Kong, Tat Chee Avenue, Kowloon, Hong Kong 999077, China; 9School of Chemistry and Physics and QUT Centre for Biomedical Technologies, Queensland University of Technology (QUT), Brisbane, QLD 4000, Australia

**Keywords:** cold atmospheric plasma, in vitro and in vivo plasma treatments, dose-dependent effects, tumor microenvironment, ROS/RNS

## Abstract

Cold atmospheric plasma (CAP) generates abundant reactive oxygen and nitrogen species (ROS and RNS, respectively) which can induce apoptosis, necrosis, and other biological responses in tumor cells. However, the frequently observed different biological responses to in vitro and in vivo CAP treatments remain poorly understood. Here, we reveal and explain plasma-generated ROS/RNS doses and immune system-related responses in a focused case study of the interactions of CAP with colon cancer cells in vitro and with the corresponding tumor in vivo. Plasma controls the biological activities of MC38 murine colon cancer cells and the involved tumor-infiltrating lymphocytes (TILs). In vitro CAP treatment causes necrosis and apoptosis in MC38 cells, which is dependent on the generated doses of intracellular and extracellular ROS/RNS. However, in vivo CAP treatment for 14 days decreases the proportion and number of tumor-infiltrating CD8^+^T cells while increasing PD-L1 and PD-1 expression in the tumors and the TILs, which promotes tumor growth in the studied C57BL/6 mice. Furthermore, the ROS/RNS levels in the tumor interstitial fluid of the CAP-treated mice are significantly lower than those in the MC38 cell culture supernatant. The results indicate that low doses of ROS/RNS derived from in vivo CAP treatment may activate the PD-1/PD-L1 signaling pathway in the tumor microenvironment and lead to the undesired tumor immune escape. Collectively, these results suggest the crucial role of the effect of doses of plasma-generated ROS and RNS, which are generally different in in vitro and in vivo treatments, and also suggest that appropriate dose adjustments are required upon translation to real-world plasma oncotherapy.

## 1. Introduction

Colon cancer is a type of cancer in the digestive tract, and according to the International Agency for Research on Cancer (IARC), it ranks third in morbidity and second in mortality, thus posing a serious threat to human health [1]. Surgery, radiotherapy, chemotherapy, molecular targeted therapy, and immunotherapy are commonly used to treat colon cancers. However, the recovery rate of colon cancer is quite low [2], and it is thus imperative to develop new treatment methods with high efficiency and fewer adverse reactions.

Reactive oxygen and nitrogen species (ROS/RNS) can induce apoptosis and necrosis in tumor cells with high selectivity, therefore drawing enormous interest in oncology [3,4]. Photodynamic therapy (PDT) [5] and sonodynamic therapy (SDT) [6] produce ROS/RNS with the assistance of sensitizers, which accumulate easily in the body and may cause toxic side effects [7]. Cold atmospheric plasma (CAP) treatment, as a type of physical therapy, does not require sensitizers, rendering it safer for cancer patients. CAP can be produced in atmospheric pressure close to room temperature, thus creating a mild plasma–tumor interface which is lethal to tumor cells, without causing much damage to surrounding tissues [8]. CAP has been demonstrated to release a variety of active ROS and RNS, including short-lived substances, such as (hydroxyl radical (OH**·**), nitric oxide (NO), atomic oxygen, ozone (O_3_), and others, as well as long-lived substances such as hydrogen peroxide (H_2_O_2_), nitrite ion (NO_2_^-^), nitrate ion (NO_3_^-^), and peroxyl groups [9]. Cancer cells exhibit a gradual increase in ROS and RNS levels, depending on their exposure time to CAP treatment. Additionally, excess ROS and RNS can cause tumor cell dysfunctions such as mitochondrial and cell membrane damage [10,11] as well as cell cycle arrest [12], which eventually lead to cell necrosis and apoptosis [13].

As we know, tumor tissues in vivo exist in a distinct tumor microenvironment (TME) mainly composed of cancer cells, myeloid cells, lymphocytes and blood vessels [14,15]. Earlier studies have reported that CAP promotes the transformation of pro-inflammatory macrophages into anti-inflammatory macrophages [16]. CAP-generated ROS/RNS increase macrophage phagocytosis by suppressing the expression of the immune checkpoint CD47, and significantly improve the recognition and killing rate of NK cells against tumor cells. [17,18]. However, there have been few studies on whether plasma can affect the functions of tumor-infiltrating T cells in TME, particularly CD8^+^ cytotoxic T lymphocytes (CTLs).

Herein, the relationship between ROS/RNS in gradient levels and apoptosis in MC38 colon cancer cells during CAP treatment is studied in vitro. CAP treatment causes necrosis and apoptosis in MC38 cells, depending on the intra- and extracellular levels of ROS/RNS. Additionally, the effects of CAP treatment on the growth of subcutaneous tumors in mice are investigated in vivo. In contrast to in vitro cell apoptosis, in vivo CAP treatment for 14 days decreases the proportion and number of tumor-infiltrating CD8^+^T cells while increasing PD-L1 and PD-1 expression in the tumors and the TILs, which promotes tumor growth in the studied C57BL/6 mice. Furthermore, the ROS/RNS levels in the tumor interstitial fluid of the CAP-treated mice are significantly lower than those in the MC38 cell culture supernatant. The findings elucidate that low doses of ROS/RNS derived from in vivo CAP treatment may activate the PD-1/PD-L1 signaling pathway in the tumor microenvironment and lead to an undesired tumor immune escape. Therefore, we propose that the effects of CAP oncotherapy are influenced by ROS/RNS levels generated in in vitro and vivo treatments, and precise dose regulations are required to achieve a greater effect of plasma oncotherapy.

## 2. Materials and Methods

### 2.1. Measurements of CAP Devices

Using the same power supply, two low-temperature plasma devices with different discharge electrodes were used in our work. For in vitro experiments, we used multiple needles as discharge electrodes. For in vivo experiments, we used a single needle as a discharge electrode due to the small operating space. (Figure 1a,b). The discharge voltage of both plasma devices was measured by a PINTECH HVP-15HF 1000:1 attenuation high-voltage probe during the discharge. Additionally, the discharge current was measured by a PINTECH PT710-D current sensor (Figure 1c,d). The optical emission spectra for both devices were recorded in the range of 320–1020 nm, using a charge-coupled device (CCD) detector (specifications: 10:100 b; Princeton instrument) placed 27 cm from the lens (Figure 1e,f). The key parameters of both CAP devices were measured in same environment and at room temperature.

### 2.2. Cell Cytotoxicity and Cell Proliferation Activity Assays

The MC38 cells were seeded in 96-well cell plate with 2 × 10^5^ cells per well in the DMEM medium (Cat# 04–001–1ACS, Biological Industries, Kibbutz Beit Haemek, Israel) at 37 °C under 5% CO_2_. The LDH and CCK-8 levels in the MC38 cells were determined for cell cytotoxicity and cell proliferation after varying the CAP treatment frequencies (once in three days, twice in three days, and thrice in three days) and durations (1 min every time, 3 min every time, and 5 min every time), and the samples were designated as on-1, 3, 5, tw-1, 3, 5, and th-1, 3, 5, respectively. Plasma kills tumor cells by generating ROS/RNS. The activity of ROS/RNS decreased with time. Increasing the frequency and duration of plasma treatment can prolong the contact time of active ROS/RNS with cells and improve the killing effect. In the LDH assay, the cells were incubated with the LDH-releasing buffer (Cat# C0016, Beyotime Biotechnology, Shanghai, China) for 1 h, and the absorbance was measured at 490 nm (OD490) after adding the LDH-detecting mix buffer. In the CCK-8 assay (Cat # C0005, Topscience, Shanghai, China), the absorbance was recorded at 450 nm (OD450) after cell incubation for 1 h with the detecting buffer. A microplate reader (Bio-Rad 550, Bio-Rad Laboratories, Hercules, CA, USA) was employed to measure the absorbance in both assays.

### 2.3. Cell Death Detection by Flow Cytometry

CAP-induced cell death was monitored by an Annexin V-FITC and PI (Propidium Iodide) apoptosis assay kit (Cat# C0016, Beyotime Biotechnology, Shanghai, China) according to the manufacturer’s instructions. In brief, 1 × 10^7^ MC38 cells were seeded in 10 mL DMEM in cell dish; after the CAP treatment, 5 × 10^5^ MC38 cells were suspended in 50 µL of PBS and stained with Annexin V-FITC and PI (Propidium Iodide) for 15 min at room temperature and in darkness. The stained cells were analyzed by flow cytometry (CytoFLEX, Beckman, Miami, FL, USA), and the data were analyzed using FlowJo software (Tree Star, Ashland, OR, USA).

### 2.4. Intracellular and Extracellular ROS/RNS Levels’ Detection

The intracellular ROS and RNS in the MC38 cells were measured by fluorescence probes DCFH-DA (Cat# S0033S, Beyotime Biotechnology, Shanghai, China) and BBoxiProbe^®^ O53 (BB-462112, Bestbio, Shanghai, China), respectively, according to the manufacturer’s instructions. Briefly, 1 × 10^7^ MC38 cells were seeded in 10 mL DMEM in a cell dish. After CAP exposure, 2 × 10^5^ MC38 cells for each assay were stained with the DCFH-DA solution or BBoxiProbe^®^ O53 solution, for 1 h at room temperature in the dark. After rinsing with PBS thrice, fluorescence was monitored at specific excitation/emission wavelengths (488/525 nm for ROS and 518/606 nm for RNS).

For extracellular ROS/RNS levels’ detection, the concentrations of H_2_O_2_ and NO, which represented the levels of extracellular ROS and RNS, were measured in the culture medium using an H_2_O_2_ assay kit (Cat#S0038, Beyotime Biotechnology, Shanghai, China) and NO assay kit (Cat# S0023, Beyotime Biotechnology, Shanghai, China), respectively, according to the manufacturer’s instructions. 2 × 10^5^ MC38 cells per well were seeded in a 96-well plate; after being treated with CAP, the cells were removed and the culture medium was collected for assays of H_2_O_2_ and NO. Since NO was unstable and rapidly metabolized into nitrite, the nitrite concentration was also measured for the NO levels.

After CAP exposure, the culture medium and TIF samples were immediately incubated with the reaction solution at room temperature for 30 min. For H_2_O_2_ measure, 20 μL culture medium or TIF samples were incubated with 100 μL hydrogen peroxide detection reagent (Cat# S0038-1, Beyotime Biotechnology, Shanghai, China) at room temperature for 30 min, followed by optical density measurement at 560 nm. For NO measure, 10 μL of the culture medium or TIF samples were incubated with reaction solution 1 (5 μL NADPH, 10 μL FAD and 5 μL nitrate reductase) at room temperature for 30 min, then incubated with reaction solution 2 (10 μL LDH and 10 μL LDH buffer) at room temperature for 30 min, then finally incubated with reaction solution 3 (50 μL Griess Reagent I and 50 μL Griess Reagent II) at room temperature for 10 min. This was followed by optical density measurement at 540 nm. The optical density (OD) and fluorescence intensity were detected using a multi-mode microplate reader (SYNERGY H1, BioTek, VT, USA).

### 2.5. Establishment of Mouse Subcutaneous Tumor Model

To explore the effects of plasma treatment on tumor development in the tumor microenvironment in detail, we chose an MC38 cell line subcutaneous tumor model on the mice’s back, in accordance with some classical references [19,20,21,22]. Eight-week-old C57BL/6 male mice with an average weight of 25 ± 2 g were used in the experiments. The mice were procured from Shanghai SLAC Laboratory Animal Co. Ltd. The C57BL/6 mice were subcutaneously injected with murine colon adenocarcinoma MC38 cell line (5 × 10^5^ cells suspended in 100 µL of the cell culture media), which was obtained from Professor Bofeng Li’s Laboratory at the University of Science and Technology of China, on the right side of the back. A total of 30 mice were used in the subcutaneous tumor model and subsequent experiments. After normal breeding for seven days, the 30 biologically independent tumor-bearing mice were selected and randomly divided into the control group (Mock) and the plasma-treated (PT) group. After injection with MC38 cells, the tumor size was monitored daily. The tumor length (a) and width (b) were measured by a Vernier caliper, and the tumor volume was calculated by V = 0.5 × (a × b^2^) formula. Then, we treated the mice with CAP treatment frequencies (fourteen times in fourteen days) and durations (3 min every time) as the “dose”.

For harvesting of tumor tissues and collection of tumor infiltration fluids (TIFs), the mice’s skin on the back was carefully cut open with surgical scissors and forceps, then the tumor tissues were cut off along the edges, and the endothelia, peritonea, and other connective tissues on the surface of the tumor tissues were eliminated. Then, the harvested tumor tissues were centrifuged at 400 *g* for 10 min at 4 °C to obtain tumor interstitial fluids (TIFs). Besides, the ROS/RNS levels of TIFs were detected by following the method of extracellular ROS/RNS levels’ detection described in 2.4.

### 2.6. Quantitative RT-PCR

The mice were euthanized after the subcutaneous tumor experiment, their spleens and tumors were extracted, and the spleen lymphocytes and tumor-infiltrating lymphocytes (TILs) were isolated, Then, the total RNA was isolated from the tumor tissues or tumor-infiltrating lymphocytes (TILs) using an RNA extraction kit (Cat# 9109, Takara, Tokyo, Japan). The isolated RNA (500 ng) was reverse transcribed into cDNA with the HiScript III RT SuperMix kit (Cat# R323–01, Vazyme, Piscataway, NJ, USA) according to the manufacturer’s instructions. A quantitative PCR was performed using the ChamQ Universal SYBR qPCR Master Mix (Cat# Q711–02, Vazyme, Piscataway, NJ, USA) in the fluorescence quantitative PCR apparatus LightCycler96 System (Roche, Basel, Switzerland). GAPDH was used as the reference gene, and the PCR primer sequences are listed in Appendix A.

### 2.7. Western Blotting

The tumor tissue (10 mg) was boiled in a Western blotting (WB) sample buffer for 30 min. The proteins in the WB samples were separated by gel electrophoresis (80 V for 30 min followed by 120 V for 1 h) and transferred to a PVDF membrane (100 V for 40 min). The proteins were incubated with a PD-L1 monoclonal antibody (Cat# 66248–1-Ig, ProteinTec Chicago, IL, USA) [23] or β-actin polyclonal antibody (Cat# 20536-1-AP, ProteinTech Chicago, IL, USA) [24] and secondary antibody (Cat# SA00001-1, ProteinTech Chicago, IL, USA) [25], and the expressions of the mouse PD-L1 and actin were analyzed using a ChemiDoc Touch Imaging System (Bio-Rad Laboratories, Hercules, CA, USA).

### 2.8. IHC, IF, and H&E Staining of Mice Tumor

Immunohistochemistry (IHC), immunofluorescence (IF), and hematoxylin and eosin (H&E) staining were conducted as described. Briefly, the subcutaneous tumor tissue was washed with flowing water for 3 h, followed by soaking in gradient alcohol solutions for dehydration. The dehydrated tumor tissue was fixed in 4% paraformaldehyde and embedded in paraffin. The tumor tissue samples were excised into 4-µm serial sections, deparaffinized with xylene, rehydrated in alcohol, and stained by H&E. In the IF staining experiment, the excised sections were incubated with the PD-L1 monoclonal antibody (Cat# 66248–1-Ig, ProteinTech Chicago, IL, USA) [26], CY3-Goat anti-Rabbit IgG (Cat# SA00009-2, ProteinTech, Chicago, IL, USA) [27] and DAPI for staining of the nuclei. Then, the samples were examined under a fluorescence microscope. In the IHC investigation, the samples were incubated with antibodies against PD-L1 (Cat# 66248–1-Ig, ProteinTech Chicago, IL, USA) [28], PD-1 (Cat# ab52587, Abcam) [29], CD3 (Cat# 17617-1-AP, ProteinTech Chicago, IL, USA) [30], CD4 (Cat# 67786-1-Ig, ProteinTech Chicago, IL, USA) [31], CD8α (Cat# ab209775, Abcam Boston, MA, USA) [32], and HRP-Goat anti-Rabbit IgG (Cat# 111-035-003, Jackson ImmunoResearch Philly, PA, USA) [33] before DAB chromogenic and hematoxylin staining. The sections were examined under an optical microscope. The antibodies used in the IF and IHC analyses are listed in Appendix A.

### 2.9. Subsets of Immune Cells Detection by Flow Cytometry

In brief, mice spleens were cut into pieces, filtered through 40 μm strainers with PBS and centrifuged for 10 min at 700 *g* before red blood cell (RBC) lysis for 5 min at room temperature; finally, they were suspended in 1 mL PBS with 10% FBS. The mice tumors were cut into pieces and shaken for 30 min at 180 r.p.m in 37 °C with 4 mL of digestive solution (DMEM containing 50 U mL^−1^ collagenase IV (Sigma, Tokyo, Japan), 20 U mL^−1^ DNase I (Cat# 69182, Sigma-Aldrich, St. Louis, MO, USA) and 10% FBS), filtered through 40 μm strainers, centrifuged for 10 min at 700 *g* with PBS before centrifugation in 5 mL 30% Percoll lymphocyte separation medium (Cat#17089109, Cytiva, Washington, WA, USA), and suspended in 1 mL PBS with 10% FBS. For flow cytometry, a 30 μM spleen lymphocyte suspension or 50μM tumor lymphocyte suspension was stained in the dark at 4 °C for 30 min by different fluorescently labeled antibodies, including CD3-PE (Cat# 100308, Biolegend San Diego, CA, USA) [34], CD19-PE (Cat# 115508, Biolegend San Diego, CA, USA) [35], CD64-PE (Cat# 558455, BD Franklin Lakes, NJ, USA) [36], Ly6G-PE (Cat# 561104, BD Franklin Lakes, NJ, USA) [37], CD69-FITC (Cat# 104505, Biolegend San Diego, CA, USA) [38], MHC II-FITC (Cat# 107605, Biolegend San Diego, CA, USA) [39], Ly6C-FITC (Cat# 128005, Biolegend San Diego, CA, USA) [40], CD8α-APC (Cat# 100711, Biolegend San Diego, CA, USA) [41], F4/80-APC (Cat# 123115, Biolegend San Diego, CA, USA) [42], Gr1-APC (Cat# 108411, Biolegend San Diego, CA, USA) [43], CD45-PE-Cy5 (Cat# 103132, Biolegend San Diego, CA, USA) [42], NK1.1-PE-Cy7 (Cat# 108713, Biolegend San Diego, CA, USA) [44], CD11c-PE-Cy7 (Cat# 117318, Biolegend San Diego, CA, USA) [45], CD3-APC-Cy7 (Cat# 101226, Biolegend San Diego, CA, USA) [34], CD11b-APC-Cy7 (Cat# 563106, Biolegend San Diego, CA, USA) [35], CD4-BV510 (Cat# 115508, BD Franklin Lakes, NJ, USA) [46], and dead cell dye. After being washed and suspended in PBS, the stained cells were analyzed by flow cytometry (CytoFLEX, Beckman, Miami, FL, USA), and the data were analyzed using FlowJo software (Tree Star, Ashland, OR, USA).

### 2.10. Co-Culture of MC38 Cells and Spleen Immune Cells

Mice spleens were cut into pieces, filtered through 40 μm strainers with PBS, and centrifuged for 10 min at 700 *g* before red blood cell (RBC) lysis for 5 min at room temperature. Finally, they were suspended in 1 mL DMEM with 10% FBS. The 1 × 10^6^ splenic lymphocytes and 1 × 10^6^ MC38 cells were mixed in 2 mL DMEM (containing 10% FBS, 1 μg mL^−1^ CD3 (Biolegend, San Diego, CA, USA) and 0.5 μg mL^−1^ CD28(Biolegend, San Diego, CA, USA)) and co-cultured in a 6-well cell plate for three days at 37 °C and in 5% CO_2_. During the co-culture experiment, splenic lymphocytes and MC38 cells were treated with CAP as described in the manuscript, and all procedures were performed under sterile conditions.

### 2.11. Statistical Analysis

A statistical method was not adopted to predetermine a suitable sample size, and data from all the mice were included. The IF and IHC experiments were blind experiments. Data from three independent experiments or mice were presented as mean ± SD. To make the comparison reliable, we have carried out normal distribution tests (Shapiro–Wilk test) of the variables using SPSS software before conducting the parametric tests. Since the sample size for each set of variables was below 50, we used the Shapiro–Wilk test for normal distribution tests, and a frequency distribution histogram of the variables was used to assist the analysis. The *p* values of each set of variables were above 0.05 (for example, the *p* values of the tumor PD-1 expression level variable in the mock and PT mice were 0.584 and 0.601, respectively), indicating that the variables basically conformed to a normal distribution. Statistical analyses were performed by either a two-tailed Student’s *t*-test to compare data with a single variable or a two-way analysis of variance to compare data with two or more variables using GraphPad (Prism, Folsom, CA, USA). Differences were considered statistically significant when *p* was <0.05.

## 3. Results

### 3.1. CAP Device for Tumor Treatment

As shown in Figure 1, two low-temperature plasma devices were designed and manufactured. The two devices used air as the work gas. The in vitro device is composed of an array of needle electrodes, a DC power supply, an oscilloscope, a high-voltage probe, a microammeter, a low-voltage probe, and a resistor (Figure 1a). When the power was switched on, the needle electrodes were connected to a high-voltage DC power supply, and plasma was generated between the copper electrodes and the external electrodes above the cell culture dish by ionizing the surrounding air. The in vivo device comprises a single needle electrode, a DC power supply, an oscilloscope, a high-voltage probe, a microammeter, a low-voltage probe, and a resistor (Figure 1b). Before the device worked, to provide enough working gas for plasma treatment, the syringe was punctured into the skin of the mouse back above the tumor mass, and 5 mL of air was injected. The syringe tip was used as the needle electrode connected to the plasma device by a copper piece. Another piece of copper was fixed to the left foot of the mouse and also connected to the ground electrode. Then, the power was switched on, and the discharge duration was 180 s for each mouse. (Figure 1b).

The key parameters of both CAP devices were measured (Figure 1c,d). In the in vitro multi-needle device, the peak-to-peak current is 1 mA, the peak-to-peak voltage is 3200 V, the amplitude is 1600 V, and the frequency is 5 kHz (Figure 1c). In the in vivo single-needle device, the peak-to-peak current is 20 mA, the peak-to-peak voltage is 1600 V, the amplitude is 1000 V and the frequency is 250 Hz (Figure 1d).

Optical emission spectroscopy (OES) is mainly used for qualitative analysis of reactive oxides and nitrogen species; the spectral peak positions are utilized for substance identification. In our work, OES is utilized to determine the composition of the reactive species generated during the CAP treatment for both needle-type devices (Figure 1e,f). A range from 354 to 425 nm indicates the molecular transitions of the N2+ second positive system [47], whereas the characteristic emission peak of atomic oxygen appears at 777 nm [48]. According to this, the emission peaks observed in the in vivo and in vitro devices (Figure 1e,f) represent an atomic oxygen radical and an ionized nitrogen molecule, respectively; thereby, all of the results demonstrating the discharge parameters of the in vivo device are comparable to those of the in vitro device, and both of the devices can produce the same kinds of reactive oxides and nitrogen species (ROS/RNS).

### 3.2. CAP Treatment Induces Cytotoxicity, Apoptosis, and Necrosis of MC38 Tumor Cells through ROS/RNS in a Dose-Dependent Manner

The in vitro CAP device is applied to MC38 cells, followed by cytotoxicity and proliferation assays (Figure 2). The intracellular LDH levels which represent the cell survival rates are inversely proportional to the cytotoxicity (Figure 2a). In addition, a CCK-8 assay is employed to evaluate the cell proliferation activity (Figure 2b). Compared to the control group, the tw-3, tw-5, th-1, th-3, and th-5 PT groups exhibit decreased survival rates in the tumor cells. The MC38 cell proliferation activity decreases dramatically in the high CAP treatment-generated ROS/RNS dose PT group, particularly the th-3 and th-5 groups. The cytotoxicity in the MC38 cells increases with a concurrent decrease in the cell proliferation activity, as the CAP treatment-generated ROS/RNS doses increase (Figure 2a,b). Notably, the cytotoxicity is higher in the th-3 group than the tw-5 group, suggesting that increasing the frequency of CAP treatment is an efficient strategy in CAP oncotherapy (Figure 2a,b).

To explore the mechanism of the decreased proliferation of MC38 cells after CAP treatment, cell apoptosis is monitored by fluorochrome FITC-conjugated Annexin-V and PI. The viable cells are not stained by Annexin V-FITC or PI, whereas the apoptotic cells are stained positive for Annexin V-FITC, but negative for PI. The necrotic cells are stained double-positive for Annexin V-FITC and PI. The proportion of apoptotic and necrotic cells is analyzed by flow cytometry (Figure 2c,d). The on-3, tw-3 and th-3 PT groups are selected for the apoptosis assay, and the results of tw-3 and th-3 PT groups reveal that the proportion of apoptotic and necrotic cells increases with increasing frequencies of CAP treatment. The results demonstrate that the CAP-induced cytotoxicity, apoptosis, and necrosis of MC38 tumor cells in vitro are CAP treatment-generated ROS/RNS dose-dependent.

As the plasma contains a variety of ROS and RNS, the levels are detected for different CAP treatment-generated ROS/RNS doses in the MC38 cell culture supernatant, using H_2_O_2_ and NO concentrations as indicators (Figure 2e,f). The nitrite concentration represents the NO content based on the NO assay kit. As expected, the concentrations of H_2_O_2_ and NO in the cell culture supernatant increase with increasing exposure times and frequencies of CAP treatment (Figure 2e,f), and the intracellular ROS and RNS levels in the th-5 group increase as well, followed by a slight decrease due to the reduction in the viable cells (Figure 2g,h). In general, in vitro CAP treatment increases both the intracellular and extracellular ROS/RNS levels in an exposure time- and frequency-dependent manner to induce cytotoxicity, apoptosis, and necrosis in MC38 tumor cells.

### 3.3. CAP Treatment In Vivo Accelerates Subcutaneous Tumor Growth in Mice

First, we confirmed that both CAP devices induce apoptosis and necrosis in the MC38 cells to the same extent (Appendix A). Then, to observe the apoptotic and necrotic effects of CAP treatment on tumor tissues in vivo, the tumor-bearing mice were treated with the in vivo CAP device (Figure 3a). The CAP treatment of the PT group mice is represented in Figure 3a. The PT group mice exhibit rapid tumor growth, and heavier and visibly larger tumors compared to the control group (Figure 3b,d). In brief, the results of the tumor experiments indicate that CAP treatment stimulates tumor growth.

### 3.4. CAP Treatment In Vivo Inhibits Expansion of Tumor-Infiltrating CD8^+^T Cells

Various immune cells are involved in limiting tumor development [49]. To explore the reason underlying the promotion of tumor growth after in vivo CAP treatment, the proportion and number of tumor-infiltrating immune cells in the spleens and tumors of the mice are determined by flow cytometry (Figure 4). In the spleen, there are no significant differences in the number and proportion of total T cells, CD4^+^T cells, B cells, NK (natural killer cells), macrophages, DCs (dendritic cells), and MDSCs (myeloid-derived suppressor cells) between the two groups (Figure 4a,b). The flow cytometry gating strategy for these immune cells was detailed in Appendix A. Interestingly, the number and proportion of tumor-infiltrating T cells and CD8^+^T cells decrease dramatically for the PT group (Figure 4c,d). Immunohistochemical staining (IHC) also demonstrates that the tumor-infiltrating CD8^+^T cells decrease significantly in the PT group (Appendix A). Since CD8^+^ cytotoxic T lymphocytes (CTLs) are key to tumor cell killing, the results indicate that the in vivo CAP treatment inhibits proliferation of tumor-infiltrating CD8^+^ CTLs, but has no effect in suppressing migration of T cells out of the spleen.

### 3.5. CAP Treatment In Vivo Activates the PD-1/PD-L1 Immune Checkpoint Signaling Pathway in Tumor Tissues

The expressions of the main immune checkpoints in the tumor tissues are quantified using RT-qPCR to determine their involvement in tumor growth post-treatment. The PD-1 expression is higher in the sorted tumor-infiltrating T cells in the PT group (Figure 5a), and at the same time, the expression of PD-L1 is higher in the tumor tissues of CAP-treated mice (Figure 5b).

Western blotting also demonstrates that tumor PD-L1 expression increases in the CAP-treated mice (Figure 5c and Appendix A). The expressions of cytokines, IFNγ, granzyme B (GZMB), and IL-2 in the sorted T cells are attenuated in the CAP-treated mice, indicating that the functions and activities of tumor-infiltrating T cells are impaired by CAP treatment (Figure 5d). Furthermore, the expression of CTLA-4 in the sorted T cells increases slightly in the CAP-treated group, in a manner similar to the expression of its ligand, CD80, in the tumor cells; however, the difference is not significant (Figure 5a,b). To determine the changes in the expression of immune checkpoint-related genes, the tumor tissue sections are subjected to IHC and IF staining. The tumor tissues from the CAP-treated mice express higher levels of PD-1 and PD-L1 than the control mice (Figure 5e). Hence, in vivo CAP treatment inhibits the expansion and effective functions of cytotoxic tumor-infiltrating CD8^+^T cells (CTLs) via elevating the PD-1/PD-L1 immune checkpoint signaling level, thereby resulting in tumor immune escape.

### 3.6. CAP Treatment In Vivo Slightly Increases ROS/RNS in the Tumor Microenvironment

As shown above, contradictory results were obtained from in vitro and in vivo experiments, and so it is postulated that the different ROS/RNS levels generated by CAP lead to different cellular responses. For verification, the concentrations of H_2_O_2_ and nitrite in the interstitial tumor fluids (TIFs) are evaluated to determine the ROS and RNS levels in the TME (Figure 6). Comparable to the control group, the concentrations of H_2_O_2_ and nitrite in the plasma-treated mice (PT group, in vivo) are significantly elevated (Figure 6a,b). The ROS/RNS levels in vitro (cell culture supernatants) and in vivo (TIFs) have been compared precisely and appropriately, and all of the measurements were taken at the same room temperature and atmospheric pressure. The concentrations of H_2_O_2_ and nitrite were approximated to the level of the on-1 (in vitro) group (Figure 2e,f). At this concentration, MC38 cells did not exhibit statistically significant damage or death. Therefore, it can be inferred that the low levels of ROS/RNS in the TME cannot inhibit tumor growth, but facilitate tumor immune escape via the PD-1/PD-L1 pathway through CD8^+^T cells.

### 3.7. High CAP Treatment-Generated ROS/RNS Dose In Vitro Induces Apoptosis in Both T Cells and Tumor Cells

The low-level ROS/RNS in TME may damage TILs cells in the tumor tissues. To compare different mortality rates between tumor cells and T cells, caused by CAP treatment, the MC38 cells are co-cultured with mouse splenic lymphocytes in vitro. The co-cultured cells are treated with three gradient exposure times and frequencies of CAP treatment (on-3, tw-3, and th-3). Flow cytometry reveals that the CAP treatment-induced death of both tumor cells and T cells is exposure time- and frequency-dependent. However, cell death is more significant in T cells, as evidenced by the greater decline in the number and proportion of T cell subsets (Figure 7a,b) and the decrease in the T cell to tumor cell ratio (Figure 7c). This implies that the tumor cells in vitro are more tolerant to CAP than the T cells. Furthermore, as the PD-1^+^ T cells show insignificant changes over the course of three days of CAP treatment (Figure 7d), it should be considered that in vitro experiment days are too short to observe changes in PD-1^+^ T cells.

The results indicate that in vitro CAP treatment induces apoptosis or necrosis in tumor cells depending on the ROS/RNS levels. However, low doses of ROS/RNS in TME derived from in vivo CAP treatment more likely accelerate the exhaustion of tumor-infiltrating CD8^+^T cells and the immune escape of tumor cells via the PD-1/PD-L1 signaling pathway. Therefore, the different doses and delivery rates of ROS/RNS between in vivo and in vitro CAP treatments may account for the contradictory results observed in the MC38 tumor cells.

## 4. Discussion

### 4.1. Different Concentrations of ROS/RNS Show Double-Sided Effects on Tumor Cells In Vivo and Vitro

An increasing number of studies show that plasma treatments may generate an interface filled with ROS/RNS which induce tumor cell apoptosis and necrosis in mice. Herein, the therapeutic effects of CAP are shown to be influenced by the levels of plasma-originated ROS/RNS. High levels of ROS/RNS induce apoptosis and necrosis in tumor cells, whereas low levels of ROS/RNS may accelerate the exhaustion of tumor-infiltrating CD8^+^T cells and the immune escape of tumor cells via the PD-1/PD-L1 signaling pathway.

Plasma therapy based on exogenous ROS has emerged as a viable strategy for tumor treatment with many advantages. For instance, plasma-generated ROS/RNS can target tumors directly, particularly in the case of skin-surface or subcutaneous tumors, possibly leading to fewer side effects compared to chemotherapy and radiotherapy.

Our results revealed that the long exposure times and high frequencies of CAP treatment in vitro effectively induce apoptosis in tumor cells due to the high levels of ROS/RNS. Our findings are consistent with other authors’ studies reporting that H_2_O_2_, NO_2_^–^, NO_3_^–^, •OH, $O_{2}^{-}$·, and singlet oxygen (^1^O) are the main components of ROS/RNS that promote apoptotic and necrosis in tumor cells [50]. Based on these arguments, H_2_O_2_ and NO (including NO_2_^–^ and NO_3_^–^) are selected as indicators of ROS/RNS levels in the present study.

Higher plasma treatment frequencies and durations decrease the activity of tumor cells. In addition, the cell toxicity activity in the th-3 PT group (thrice in three days, 3 min every time) shows no significant difference compared to the th-5 PT group (thrice in three days, 5 min every time). Therefore, in in vivo experiments, the mice are treated with CAP for 3 min daily.

The anti-cancer mechanisms of CAP remain poorly understood. Defining the anti-cancer mechanisms of CAP continues to be a key challenge in this field; ROS/RNS-induced oxidative damage in cancer cells is emerging as one of the most probable underlying mechanisms [51].

Plasmas may deliver ROS/RNS only 150 µm–3.0 mm below the tumor surface, suggesting that the effective penetration of gaseous plasmas is no greater than 3 mm [52]. Therefore, in the present study, to avoid interferences from mice skin regarding the delivery efficiency of CAP-derived ROS/RNS, a small volume of air is subcutaneously injected into the mouse body. This is followed by ROS/RNS generation by the in vivo CAP device. The key characteristics and parameters, such as the voltage, current waveform, optical emission spectra and gaseous CAP yield, are evaluated and compared between the in vivo and in vitro CAP groups (Figure 1). In addition, induction of in vitro MC38 cell death by both devices is analyzed (Figure 2 and Appendix A). The results confirm that both CAP devices induced cell death. However, the in vivo experiments reveal that contrary to induction of cell apoptosis and necrosis in vitro, tumor suppression is not achieved in the plasma–tissue interface.

Furthermore, the ROS/RNS levels in TME are significantly lower in vivo than in vitro, the former being equivalent to the levels in the on-1 and on-3 groups. The low level of ROS causes minimum cytotoxicity to the tumor cells, as evidenced by the cell proliferation assay (Figure 1). ROS/RNS have been shown to impose double-sided effects on tumor cells depending on the concentration [53]. In general, low concentrations of ROS/RNS can induce proliferation and differentiation [54]. As the concentration increases, ROS/RNS can damage cellular biomolecules and cause gene mutation to promote tumor genesis, development, and metastasis [55]. ROS/RNS have been reported to induce apoptosis, necrosis, and immunogenic cell death (ICD) of tumor cells when certain threshold concentrations are exceeded [5,6]. In accordance with these findings, our results show that low levels of ROS/RNS promote cell proliferation at threshold concentrations, corresponding to different effects of plasma on cells. Moreover, differences in ROS/RNS levels in tumors during CAP treatment between the in vivo CAP device used in the present study and equipment used in other studies can explain the differences in tumor growth.

### 4.2. The Plasma Affects the Biological Activities of the TILs in the Tumor Microenvironment

The tumor microenvironment, which contains various immune cells, blood vessels, and an extracellular matrix, is highly intricate when compared to the culture medium; this benefits tumor cell proliferation [56]. Therefore, it can be inferred that CAP treatment in vivo affects certain immune cell populations. Previous studies have indicated that relatively high levels of ROS in the TME can affect the activities of tumor-infiltrating lymphocytes (TILs) and limit their anti-tumor effects [57]. Siska et al. have demonstrated that CD8^+^ TILs in renal clear cell carcinomas are functionally impaired due to excessive mitochondrial ROS, and that the CD8^+^ TILs are rescued upon treatment with mitochondrial ROS scavengers [58]. Goodwin et al. have observed that increased levels of ROS originating from bacterial polyamine catabolic pathways lead to tumor formation in the colon [59]. A study on T cells expressing chimeric antigen receptor (CAR-T) revealed that upregulation of CAR-T is correlated with decreased intracellular oxidative stress. In accordance with these findings, the present study reveals that CAP treatment attenuates CD8^+^ TILs without affecting splenic T cells. The quantities of CD4^+^T cells remain unchanged in the tumors of PT mice, while the CD8^+^T cells decrease. During tumor growth, TILs such as NK cells and CD8^+^T cells (cytotoxic T lymphocyte, CTL) are responsible for tumor killing, while some cells inhibit tumor immunity, including CD4^+^ regulatory T cells (Treg, the main composition of tumor CD4^+^T cells), myeloid-derived uppressor cells (MDSC) and tumor-associated macrophages (TAM) [14,15]. These immunosuppressive cells are tolerant to a TME with qualities such as high ROS levels an a low pH, and help tumors to achieve immune escape [60]. Compared with CD4^+^ Tregs, CD8^+^ CTLs and other anti-tumor cells have lower adaptability to the TME, are more vulnerable to high ROS and other factors, and become exhausted during tumor development [61]. In our in vivo study, slightly elevated ROS levels in the plasma-treated tumors caused exhaustion of CD8^+^T cells. The CD4^+^T cells, on the other hand, do not demonstrate a reduction, probably due to tolerance to high ROS.

### 4.3. Low Doses of ROS/RNS Derived from In Vivo CAP Treatment May Lead to Tumor Immune Escape via the PD-1/PD-L1 Signaling Pathway

It has been shown that the CAP treatment involves regulation of the tumor immune checkpoints. Abraham et al. have reported that the innate immune checkpoint CD47 present on tumor cells is oxidized by CAP treatment in vitro as well as in vivo, resulting in reduced tumor immunosuppression [18]. Ramona et al. have demonstrated that CAP amplifies the natural killer (NK) cell cytotoxicity in skin cancer mouse models [17]. However, the present study reveals that CAP treatment enhances the expression of PD-1 and PD-L1 in T cells and tumor cells, respectively, but reduces the expression of T cell cytokines (IFNγ, GZMB, and IL-2) that are associated with CTL functions and activity. This indicates that low levels of ROS/RNS derived from CAP treatment activate immune checkpoint signaling. Moreover, tumor size is positively correlated with the expressions of PD-1, PD-L1, and T cell cytokines, and negatively correlated with the ROS/RNS levels. Tumor cells have been reported to evade immune surveillance by immune checkpoint signaling activation, for example, through PD-1/PD-L1 activation, and ROS/RNS removal improves the therapeutic effects of the PD-1/PD-L1 checkpoint blockade [62]. In addition, tumor cells treated with metformin to decrease mitochondrial ROS/RNS levels exhibit activation of tumor T cells and enhanced therapeutic effects of the PD-1 blockade [63,64]. Maj et al. have reported increased ROS and oxidative stress in the TME, which ultimately promotes the immunosuppression effect of Tregs and eliminates the therapeutic efficacy of the PD-L1 blockade in mice with tumors [65]. These findings suggest that a certain concentration range of ROS/RNS enhances PD-1/PD-L1 signaling through different mechanisms.

To further compare the effects of CAP treatment on immune and tumor cells cultured under the same conditions, in vitro co-culture experiments are conducted. The decline in T cells is more significant than that in MC38 cells upon CAP treatment, implying that T cells are more sensitive to plasmas than tumor cells. In addition, a slight increase is observed in the PD-1^+^ T cells, although the difference is insignificant relative to the tumor PD-1^+^ T cells. This can be attributed to the short incubation duration in vitro. Another possible explanation is that the proliferation ability of in vitro splenic T cells is much slower than that of MC38 cells, and the in vitro medium cannot accurately simulate the complex microenvironment of tumor tissues. The results suggest that CAP therapy has to be optimized to maximize the effectiveness of the targeted delivery of ROS/RNS.

## 5. Conclusions

Our study reveals that the effects of CAP oncotherapy are influenced by ROS/RNS levels. High concentrations of ROS/RNS derived from CAP result in cytotoxicity to tumor cells in vitro, but in the in vivo CAP treatment, a slight increase in ROS/RNS in the TME promotes activation of immune checkpoint signaling. This occurs through upregulation of PD-1/PD-L1 expression, which consequently leads to the exhaustion of tumor-infiltrating CD8^+^ CTLs. The novelty of the present study is that the effect of doses of plasma-generated ROS/RNS in in vivo treatment are generally different from those in in vitro treatments. The results also reveal potential risks due to the complex tumor microenvironment of in vivo CAP treatment. In vivo, certain doses of plasma-derived ROS/RNS can reshape the tumor microenvironment, which consequently influences activation of immune checkpoints and induces exhaustion of tumor-infiltrating T cells. Based on the subcutaneous tumor model study, we will refine the device for intestinal treatment in the study to then explore the effects of plasma on in situ spontaneous colon tumors in mice.

Therefore, future investigations must focus on precisely regulating and monitoring ROS/RNS levels to activate ROS/RNS-induced apoptosis signaling in tumor cells, in order to achieve real-world plasma oncotherapy.

## Figures and Tables

**Figure 1 biomolecules-13-00707-f001:**
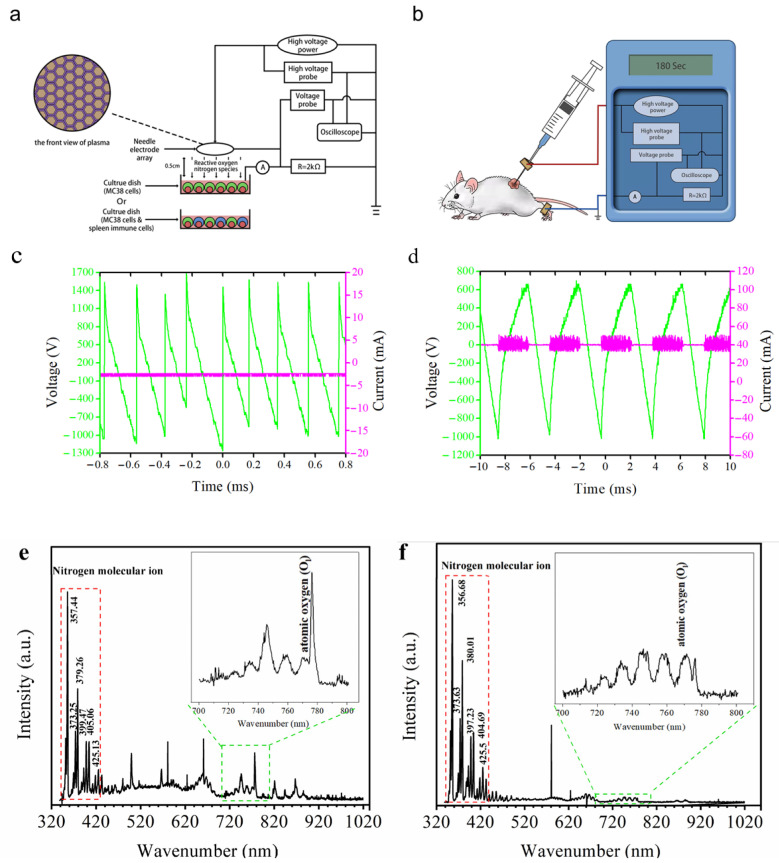
Schematic diagrams and characteristic parameters of the cold atmospheric plasma (CAP) devices for tumor treatment in vitro and in vivo. The schematic diagrams of the CAP devices for treating cells and mice are shown. (**a**) The in vitro device comprises an array of needle electrodes, a DC power supply, an oscilloscope, a high-voltage probe, a microammeter, a low-voltage probe, and a resistor. (**b**) The syringe is punctured into the skin of the mouse back above the tumor mass, and 5 mL of air is injected as the working gas. The syringe tip is used as the needle electrode connected to the plasma device by a copper piece. Another piece of copper is fixed to the left foot of the mouse and also connected to the ground electrode. The discharge duration is 180 s. (**c**,**d**) Voltage and current waveform of the multi-needle (**c**) and single-needle (**d**) devices. (**e**,**f**) Optical emission spectra of the multi-needle (**e**) and single-needle (**f**) plasma in the wavelength range of 340–1000 nm.

**Figure 2 biomolecules-13-00707-f002:**
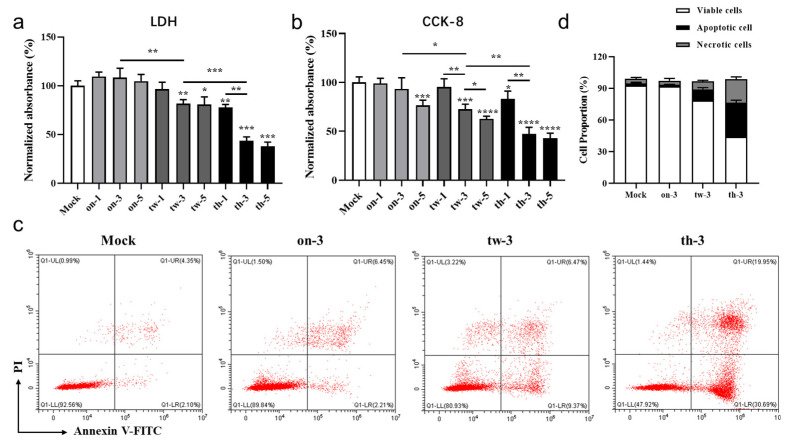
CAP treatment-induced cytotoxicity, apoptosis, and necrosis in MC38 cells in vitro via ROS/RNS generation. The MC38 cells are cultured for three days. Nine different exposure times and frequencies of CAP treatment are generated based on the different combinations of treatment frequencies and durations per treatment. (**a**,**b**) The normalized absorbance values of LDH (OD490) and CCK-8 (OD450) assays conducted with MC38 cells in both the control (mock) and PT groups, *n* = 4. (**c**,**d**) Flow cytometry analyses based on Annexin-V/PI staining to determine apoptotic and necrotic MC38 cells in the mock and PT groups (on-3, tw-3, and th-3), with logarithmic plots (**c**) and percentages of apoptotic cells (**d**), *n* = 3. (**e**,**f**) H_2_O_2_ and Nitrite concentrations in the MC38 cell culture supernatants in the mock and PT groups, *n* = 5. (**g**,**h**) Normalized fluorescence results for the levels of (**g**) intracellular ROS and (**h**) RNS in the MC38 cells treated with different CAP doses, *n* = 5. The data represent a minimum of three independent experiments (means ± SD). Statistical analysis is performed for all the PT groups using two-sided unpaired *t*-tests relative to the mock group (* *p* < 0.05, ** *p* < 0.01, *** *p* < 0.001, and **** *p* < 0.0001).

**Figure 3 biomolecules-13-00707-f003:**
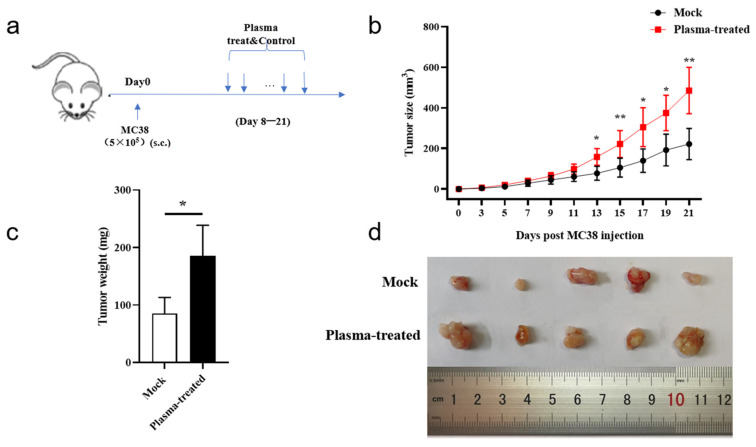
CAP treatment in vivo facilitates tumor growth in mice. (**a**) Schematic diagram depicting the tumor-bearing mice and CAP treatment. The mice are subcutaneously injected with 5 × 10^5^ MC38 cells on day 0 and then subjected to CAP treatment from days 8 to 21. (**b**) Growth curves of the subcutaneous tumors in the mock and PT groups, *n* = 5. (**c**) Weight of subcutaneous tumors in the mock and PT groups, *n* = 5. (**d**) Images of the tumors on day 21 after implantation of tumor cells. The data are from 15 biologically independent animals (mean ± SD for (**b**,**c**)) and the statistical analysis is performed using two-sided unpaired *t*-tests (* *p* < 0.05 and ** *p* < 0.01).

**Figure 4 biomolecules-13-00707-f004:**
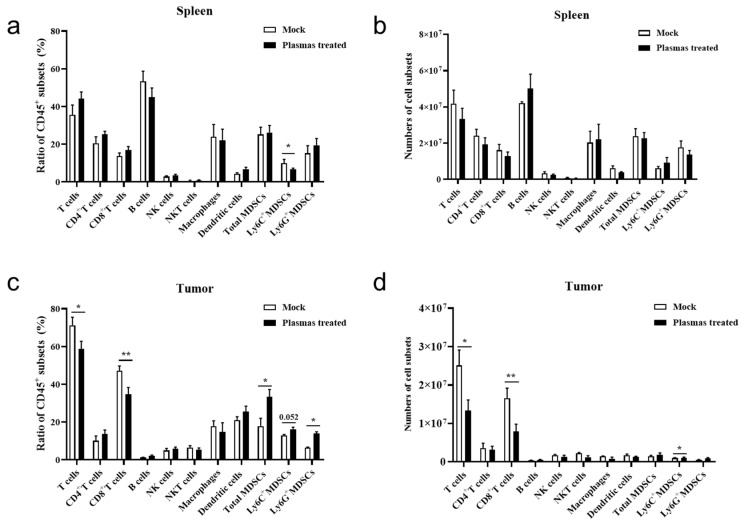
CAP treatment in vivo inhibits expansion of tumor-infiltrating CD8^+^T cells. (**a**,**b**) Number and proportion of splenic immune cell subsets and (**c**,**d**) Tumor-infiltrating immune cell subsets in the mock and PT group mice determined by flow cytometry, *n* = 5. The data are from five biologically independent animals (means ± SD) and the statistical analysis is performed using two-sided unpaired *t*-tests (* *p* < 0.05, and ** *p* < 0.01).

**Figure 5 biomolecules-13-00707-f005:**
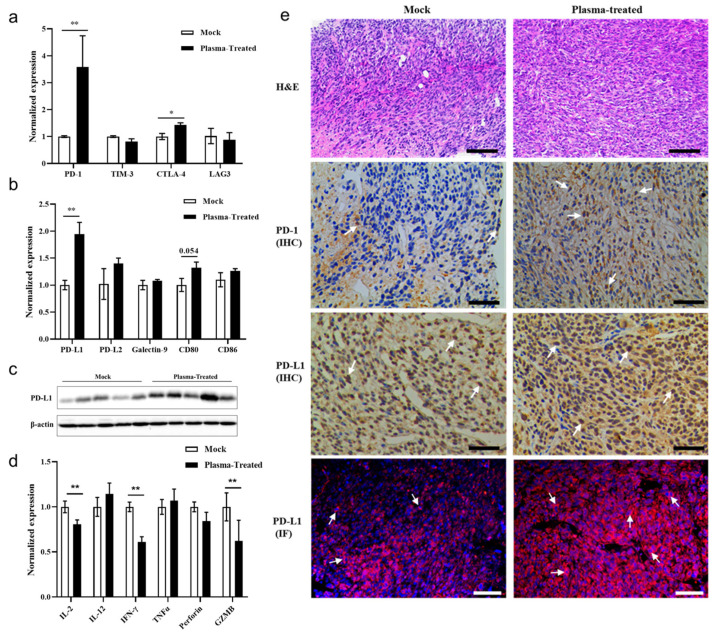
CAP treatment in vivo activating the PD-1/PD-L1 immune checkpoint signaling pathway in tumor tissues. (**a**) RT-qPCR analysis to determine the expressions of co-inhibitory receptor genes (PD-1, TIM-3, CTLA-4, and LAG-3) in the tumor-infiltrating T cells sorted from the tumors of the control and PT groups, *n* = 5. (**b**) RT-qPCR analysis to determine the expressions of co-inhibitory ligand genes (PD-L1, PD-L2, Galectin-9, CD80, and CD86) in the tumor tissues from the control and PT group, *n* = 5. (**c**) Immunoblot analysis to determine the PD-L1 expression in the tumor tissues of the control and PT group mice, *n* = 5. (**d**) RT-qPCR analysis to determine the cytokine expressions in the tumor-infiltrating T cells sorted from the tumors of the control and PT groups, *n* = 5. (**e**) Immunohistochemical (IHC) staining for PD-1 and PD-L1, and immunofluorescence (IF) for PD-L1 in the tumor tissues from the control and PT groups. The arrows indicate the PD-1^+^ or PD-L1^+^ cells. Scale bars are 100 µm, *n* = 5. The data are from ten biologically independent animals (means ± SD for (**a**,**b**)) and the statistical analysis is performed using two-sided unpaired *t*-tests (* *p* < 0.05 and ** *p* < 0.01).

**Figure 6 biomolecules-13-00707-f006:**
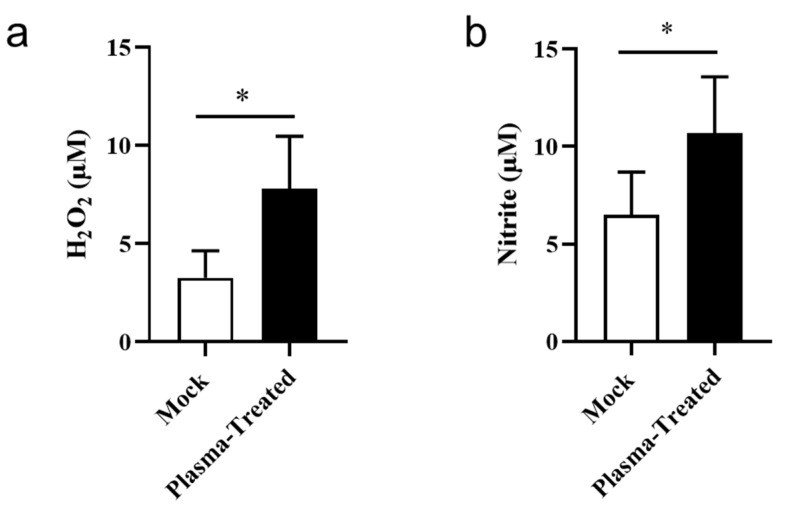
CAP treatment for 3 min daily for 14 days slightly increases ROS/RNS levels in the tumor microenvironment. Concentrations of (**a**) H_2_O_2_ and (**b**) nitrite in the tumor interstitial fluids (TIFs) in the mock and PT groups, *n* = 5. The data are from five biologically independent animals (means ± SD) and the statistical analysis is carried out using two-sided unpaired *t*-tests (* *p* < 0.05).

**Figure 7 biomolecules-13-00707-f007:**
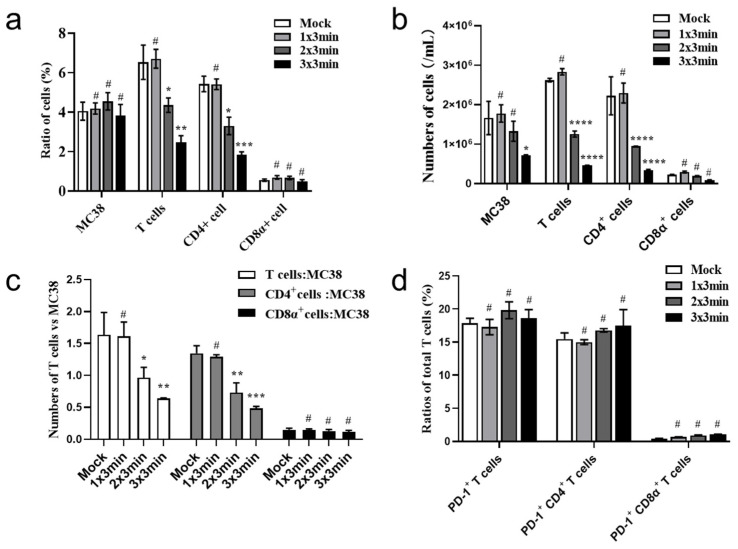
A high CAP treatment-generated ROS/RNS dose in vitro suppresses T cells more than tumor cells. The MC38 cells and splenic lymphocytes are cultured in DMEM containing 10% FBS, 1% penicillin-streptomycin, 3 µg/mL CD3, and 2 µg/mL CD28, and then treated with different exposure times and frequencies of CAP treatment (on-3, tw-3, and th-3), as illustrated in Figure 1a. (**b**) Flow cytometry analysis to determine the number and proportion of MC38 cells and T cell subsets in the mock and PT groups, *n* = 3. (**c**) Ratio of T cell subsets to MC38 cells determined based on the results in (**a**,**d**). Flow cytometry analysis of the proportion of PD-1^+^ T cell subsets in the total T cell populations, *n* = 3. The data are representative of three independent experiments (mean ± SD). The statistical analysis is performed using two-sided unpaired *t*-tests relative to the mock group (# insignificant, * *p* < 0.05, ** *p* < 0.01, *** *p* < 0.001, and **** *p* < 0.0001).

## Data Availability

Not applicable.

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
