# Peer review of "Dose-Dependent Effects in Plasma Oncotherapy: Critical In Vivo Immune Responses Missed by In Vitro Studies"

_biomolecules, 2023, doi:10.3390/biom13040707_

Round 1
Reviewer 1 Report
Review for biomolecules-2241539
In the present manuscript, the authors explore the dichotomy observed between in vitro and in vivo studies when cell or animal models are exposed to cold atmospheric plasma (CAP). This research is of high relevance, considering it respects to an emergent possible anticancer therapy, whose effects are not completely understood. This understanding is needed to achieve real-world plasma oncotherapy, which justifies the high pertinence and scientific soundness of this study.
In general, the submitted manuscript must be improved in terms of structure (mainly in methodology) and writing. The scientific rational is appropriate, but the experimental design has some flaws. Introduction provides important and updated theoretical considerations of the different topics necessary for the all understanding of the paper and a good integration of the knowledge. The discussion/conclusions are not completely sustained by the results obtained.
I think that a summary in the form of a graphical abstract will help the readers to easily understand the main ideas that authors wanted to cover.
However, I have major concerns to point out, that are then detailed below.
1. Methodology does not provide enough details, nor is clearly described. The in vitro and in vivo measurements are mixed and it is difficult to follow what was done in mice samples or just in cells. I suggest to reorganize this section, first describing the in vitro studies and then the in vivo studies. Statistical analysis also lacks rigor.
2. There are results that seem to be overstated.
In accordance with the previously mentioned, authors should provide clarifications for the following points.
1. Please correct several typos, mainly with superscript/underscripts.
2. Line 61. Please correct the molecular formula of hydroxyl radical. The formula used refers to hydroxide ion and not to the radical.
3. Line 64. Please clarify to which kind of gradual increase you are referring to. Is it an increase over time (after exposure)? Or is it an increase dependent on the exposure time to CAP (i.e. time during which the cells are exposed)?
4. Lines 72-74. Please rephrase.
5. Line 84. The authors often refer to plasma doses. What do you mean with plasma doses? CAP treatment is based on the exposure time and not doses. The term “dose” does not apply to plasma treatment. Please correct this during the all manuscript.
Concerning this, when you refer to “low doses”, do you mean “shorter exposure times”?
6. Line 91. Do you consider that a mouse subcutaneous tumor model is adequate to study tumor microenvironment? In fact, in this model, tumor develops far from the origin organ (in this case, colon) not mimicking colon tumor microenvironment.
7. Line 94. Please specify the site of tumor cells administration (back, armpit, or other…), the characteristics of cell line (supplier, passage number, origin species).
8. Line 94. Typo: “5 × 105” instead of “5 × 105”.
9. Line 99. Typo: “V = 0.5× (a× b2)” instead of “V = 0.5× (a× b2)”.
10. Figure 1a presents a co-culture of MC38 cells and spleen immune cells, however, this approach is never described in methods. Please clarify!
11. Line 119. Why did you chose a discharge duration of 180s? And why do you consider this a “low dose”? Please address this in the manuscript.
12. Line 123: Please provide more details on the methodology (nº of cells seeded, multiwells plates used, 96-well or other, etc..) or provide a previous reference with the detailed methodology
13. Lines 135, 143, 161: Please detail the nº of cells used in each assay.
14. Line 156. Please provide details on how tumor tissues were harvested before centrifugation (mechanically, using trypsin, collagenase, or other).
15. Line 158. It is not clear which reaction solution is this. Did you measured ROS and RNS in TIFs or just one of these?
16. Please provide references and suppliers for all the antibodies used.
17. Lines 194-199. Why did you just use parametric tests? Have you done an analysis of normality distribution of the variables? Parametric tests should be used just in case of normality distribution. Please clarify.
18. Line 204. Words missing.
19. Lines 215-219. The emission peaks obtained for the atomic oxygen are not quantified and are quite different in fig 1e compared to 1f. How can you guarantee that those are comparable? Please provide quantification of these peaks as provided for nitrogen.
20. Line 225, 227. CCK-8 assay does not evaluate cell counts. Please rephrase.
21. What was the rational of using different frequencies? Which differences did you expect? Please explain this rational in the manuscript.
22. Line 234. Annexin V is not a fluorochrome. The fluorochrome is FITC, which is conjugated to annexin. Please correct.
23. Lines 239-240. This sentence is not accurate. The proportion of apoptotic and necrotic cells does not increase substantially after the CAP treatment, but instead with increasing frequencies of CAP. In the on-3 condition, cell death is not observed. Please rephrase.
24. Line 241. This dependency is with the frequency. No dose exists here. And in the cell death studies you didn’t evaluated the dependency on exposure time, because you just selected the time 3 min.
25. Lines 242-251. These measurements were made immediately after the collection of the supernatants? Or were the samples immediately frozen? If not, RONS will rapidly convert and the measurements will not correspond to the real levels. Please clarify whether this was taken into consideration.
26. Lines 249-250. Did you perform comparisons between condition groups or just with control? If not, you cannot infer that a dose-dependency exists. Please clarify.
27. Line 303. Comparisons where neither # nor * appear mean that no statistical analysis was done? Please clarify.
28. Line 338 and Fig 6. The RONS increase has statistical significance, so it is not a slight increase. Please rephrase in the manuscript.
Moreover, it is not clear nor stated in the methodology, which “dose” was used for plasma treatment in this experiments of RONS measurement in TIFs. A proper and precise comparison between in vitro and in vivo experiments should be done, for the same conditions.
29. Lines 360-361. Since your variable is not the time, referring “fast decline” is not correct. You didn’t evaluate the effect over time. Please rephrase here and in the discussion section.
30. Lines 396-399. The authors cite reference 22 as their own work, when it belongs to another author.
31. Lines 402-403. It was not clear in methodology that animals were treated with CAP daily. How will these results be comparable with in vitro studies where the cells are treated once?
32. Lines 417-419. Are the conditions here compared the same? The number, time, duration of the irradiations were the same for the in vitro and vivo experiments, allowing these conclusions?
33. Lines 429-433. It is not clear why the authors selected conditions inducing low levels of ROS, considering that it is well known from the literature that low levels of ROS have a proliferative effect. The desired effect in plasma therapy is to have a high amount of ROS. Please clarify.
Overall, this experimental work is of high pertinence in the field of plasma therapy. It is imperative and urgent to clarify whether the good antitumor effects observed in vitro are clearly translated in an effective but also safe therapy in vivo. However, I consider that this work has several flaws in terms of methodology and writing, which question the guarantee of the correct interpretation of the results. For these reasons, this work is not worth of publication in Biomedicines in the present form, so major corrections must be done.
Reviewer 2 Report
In this manuscript Yuanyuan He, et al. described results of their work aimed at evaluation of antitumor effects of cold atmospheric plasma (CAP) on colon cancer cells using in vitro and in vivo models and specially designed installations for plasma generation in these models. It is known that CAP treatment induces apoptosis and necrosis in tumor cells due to the high content of reactive oxygen and nitrogen species. However, the mechanisms of CAP interaction with cancer tissue and tumor-infiltrating immune cells are still poorly understood.
In this work, the authors demonstrated a distinction in kind of the effect of plasma on tumor growth in in vitro and in vivo models. It turned out that in a mouse model of subcutaneous tumor growth, plasma treatment did not suppress, but enhanced tumor growth. This is an unexpected but very significant result of this work. The authors reasonable attribute the observed plasma effect in the in vivo model to a low dose of ROS/RNS reaching tumor cells and cells of tumor microenvironment. The results of the work indicate that such low doses of ROS/RNS can enhance the ability of tumor tissue to avoid the antitumor immune response. There is no doubt that the data presented in the manuscript will be useful for developing methods for the practical use of CAP in oncotherapy.
In general the study is well presented, proper controls are used and the conclusions are convincingly supported by experimental results, the data are of considerable novelty and interest. Manuscript is well written.
I have some minor comments:
1. The number of mice used in the experiments should be indicated in the paragraph 2.1 of Materials and Methods.
2. (line 143) RONS should be replaced by ROS and RNS.
3. (line 204) There is no In before in vivo.
4. (line 288) “In the spleen” should be inserted before “There are no …”
5. In paragraph 3.5 there are results for “sorted T cells”, but there is no information about cell sorting in Materials and Methods.
6. (line 402) There is no In before in vivo.
7. The manuscript contains a number of inaccuracies and misprints in the text. Authors should find and correct the mistakes.
Reviewer 3 Report
This is an interesting paper, which gives important information about a crucial issue concerning the use of cold plasmas for cancer treatment: the authors show that the effectiveness observed in vitro by many authors can be counteracted in vivo by immunological effects which can actually improve tumor survival and help its growth, due to exhaustion of T-cells, if plasma "dose" is not appropriate. I regard this as an important observation, which definitely deserves publication.
Before granting publication permission, a minor revision of the paper is required, according to the points listed below.
1) Line 92: Please specify how many mice were used in the experiment.
2) Line 94: I suppose that 105 actually means 10 to the power of 5. Please use appropriate superscript notation for power.
3) Line 101: Please avoid the expression "cutting edge" (also in other places in the manuscript), which sounds more as an advertisement than as a scientific term.
4) Line 104: When you write "both plasma generators", the reader suddendly discovers that there are two plasma generators. Please introduce this concept before, explaining that two plasma sources were used, one for in-vitro studies and the other for in-vivo ones.
5) The two plasma generators should be described in more detail from the point of view of the mechanical construction, so that other authors can repeat the same measurements. Furthermore, the peak voltage and current and waveforms, which are given in fig.1, should be described in the main text, giving amplitude and frequency.
6) Line 127: It took me some time to understand that frequency, in this context, means "treatment frequency". Please be more clear, since "frequency" could also refer to the frequency of the waveform used to power the plasma source.
7) Line 213: Why do you refer to "absorbance" while speaking about emission spectroscopy?
8) Line 216: "Ionized nitrogen and oxgyen" is misleading. While N2+ is indeed ionized, O is a neutral radical, while the expression could lead to think that you are talking about ionized oxygen. Please, be more precise.
9) Lines 217-219: In my view this statemeent is too strong. Actually, deriving discharge parameters from line intensities is a delicate process, and it is difficult to draw conclusions if a collisional-radiative model is not used. Furthermore, I noticed that the 777 nm O line is quite different for the two sources, suggesting that different amounts of this radical are produced.
10) Lines 344-345: I could not understand the meaning of this sentence, please rewrite it.
11) Figure 6: Please specify in the caption to which treatment duration these graphs refer to.
12) Lines 366-367: I have problems in understanding the process by which it was deduced from the results that the in-vitro treatment produces an interface. Actually, I am not even sure about what the authors actually mean by "Interface". I think that this point should be expanded and better explained.
13) Lines 410-412: The procedure of injecting a small volume of air is introduced here, but it is not clear why this is done, what is the expected beneficial effect, and how this could affect the whole procedure. Please, clarify these points.
